# Prioritizing Business Quality Improvement of Fresh Agri-Food SMEs through Open Innovation to Survive the Pandemic: A QFD-Based Model

**Tutur Wicaksono** [1,*] **, Md Billal Hossain** [1] **and Csaba Bálint Illés** [2]

1   Doctoral School of Economic and Regional Sciences, Hungarian University of Agriculture and Life Sciences, 2100 Gödöllő, Hungary; shohan_bd13@yahoo.com

2   Institute of Economic Sciences, Hungarian University of Agriculture and Life Sciences, 2100 Gödöllő, Hungary; Illes.Balint.Csaba@uni-mate.hu

*   Correspondence: tuturwicaksono18@gmail.com or Tutur@hallgato.uni-szie.hu

**Abstract:** It is important that SMEs are able to prioritize business quality by identifying business requirements based on customer requirements. This strategy is able to help SMEs generate innovation in the form of improved business quality activities to meet customer requirements. This approach uses the "quality function deployment (QFD)" method to identify the priority business requirements and improvement actions to generate business quality. As a result, we managed to identify five priority variable business requirements (BReqs) based on seven variable customer requirements (CReqs) with the lowest satisfaction score. We proposed some improvement actions in perspective quality based on five priority business requirements. Moreover, the final quality matrix of business quality improvement priority generally makes it easier for users to read and find out which variables need to be improved. This research also presents a solution so that users can perform business actions effectively and efficiently in allocating their resources.

**Keywords:** SMEs; business quality; QFD; satisfaction; improvement; agri-food

## 1. Introduction

Currently, the COVID-19 pandemic is having a devastating impact on the small and medium enterprise (SME) sector. Declining business activities are affecting the falling incomes of SMEs compared to the years before the pandemic. To date, there are no effective government measures to improve the economic conditions of the SME sector, which have been worsened by the pandemic COVID-19 [1,2].

It is important for SMEs to be innovative in their business processes so that they can survive in the current pandemic. Based on the rectangular compass conceptual model of open innovation perspective by Yun and Zhao (2020) [3], it is stated that a deep understanding of customer needs is necessary for the innovation of a customer-centric company, especially at a time when technology is rapidly evolving and influencing the dynamics of customer needs, as is the case today. Identifying customer needs and translating them into solutions for business needs can increase SME productivity. The application of open innovation promotes the flow of information that can be used by SMEs to accelerate innovation, help SMEs to adapt to market demands and make improvements to improve the quality of the business [4]. Prasanna et al. (2019) [5] noted that there are always changes in customer preferences and needs that force SMEs to constantly innovate in their business to compete with their rivals. This becomes even more important when the spread of SMEs is concentrated not only in big cities but also in rural areas, which can also affect economic development in rural areas [6,7].

Agri-food businesses are in one of the most demanding sectors of the economy. They have to compete with other businesses, anticipate logistical risks that can affect product quality, and understand market preferences and needs in order to retain their

customers [8,9]. However, Central European SMEs are not satisfied with the regulations and the role of the governments in Central Europe, which still do not provide optimal support to the agri-food industry [10]. Moreover, the quality of human resources managing agri-food in Central Europe is still considered lower than in other EU countries, making agri-food SMEs in Central Europe very vulnerable with threats to their to survival [11].

It is therefore important that SMEs in the agri-food sector are able to survive and compete in the industry. To be competitive, SMEs must use their limited resources effectively and efficiently. Effective and efficient use of resources is a factor that promotes the survival of SMEs [12–14]. In order to adequately implement this strategy, it is important for small and medium enterprises in the agri-food industry to understand their customers from a quality perspective, starting with identifying customer needs. The knowledge of customer needs can be used by the companies to find out what actions the SMEs can take to improve and meet the customer needs [15,16].

Business processes that are able to meet customer requirements lead to customer satisfaction and retain the customer [17,18]. This makes the product easier to absorb by the market, creates sustainable revenue and has an impact on the business continuity of the SME itself [19,20]. This opinion is also supported by statements of several other researchers who stated that the success of SMEs in implementing the right business strategy has brought their enterprise to an established business stage [21,22]. SMEs in Central Europe have an important role in the economic system in Europe and have managed to create new jobs that have an impact on reducing unemployment [23,24]. Even if they are not effectively managed, they have managed to stabilize the economy through their activities, especially when a country's economic activity is declining [25]. That is why the EU Government is providing financial support to SMEs so that they can continue to innovate and develop their businesses [26].

The objectives of this research are as follows. First, to identify the priority level of customer requirements that need to be improved; second, to identify the priority level of business requirements that need to be improved; and third, to identify the required improvement actions of SMEs in the agri-food market, in order to generate business quality by meeting the priority requirements of customers.

We used the quality function deployment (QFD) method, which is a customer-centric approach that is able to translate customer requirements into business processes of the enterprise. In its application, QFD has been successfully used by many researchers in various fields. Park et al. (2021) [27] have applied QFD as an effective method for designing and managing self-service technology (SST). Wu et al. (2018) [28] applied the QFD method to identify service failures and service recovery actions in the hospitality industry. Pandey (2020) [29] has applied the QFD method with the 'house of quality' as an effective tool to evaluate the strategic design parameters of an airport by integrating it with the necessary requirements. Liu et al. (2020 [30] applied a QFD-based approach to help manufacturing companies develop supplier evaluation models to create an optimal anti-counterfeiting platform. Avikal et al. (2018) [31] have applied QFD as a method to develop an approach that can help product designers easily classify customer needs for esthetic attributes in SUV product designs. QFD has also been successfully used to determine the priority of maintenance unit strategies in the aviation industry [32]. However, the QFD approach has never been used by research related to the application of business quality in the agri-food industry. This study aims to fill this gap and apply the QFD method to prioritize enterprise quality improvement in the mid-sized agri-food industry.

We adopted the QFD method by making some adjustments to the house of quality (HOQ) matrix. We eliminated several matrices to be more efficient for the purpose of this study. The expert team participated extensively in translating and determining the relationship to each term and variable used in this study. We propose several priority improvement actions that can have a significant impact and meet the priority requirement of customers as a result from the quality perspective.

Following this introduction, this article presents a literature review to contextualize the characteristics and conditions of agri-food SMEs, particularly in the pandemic period (Section 2.1). Subsequently, Sections 2.2 and 2.3 contextualize the importance of implementing business quality in SMEs and quality function deployment as the most appropriate method for delivering business quality. Section 3 describes the methodological-logical framework of this research to analyze the business quality and improvement activities of agri-food SMEs in the market in Central Europe during the pandemic. Section 4 presents the results and discussion. Conclusion, implications, limitations and future research are presented in Section 5.

## 2. Literature Review

### 2.1. Agri-Food SME Characteristics and Open Innovation

The SME sector of the agri-food industry has a specific industrial character. For example, there are strict environmental regulations, mature industrial conditions and government subsidy policies for this industry [33]. The agri-food industry is dominated by small family farms where management and control of the business are not separated. Their business processes are driven by economic goals based on strong family business traditions [34].

This industry also has product characteristics that are different from other industries, as the agri-food industry is dominated by fresh product commodities that have perishable characteristics [35,36]. This is confirmed by Perlman et al. (2019) [37], who state that the value of agri-food products can decrease due to damage and deterioration processes. Not only that, but fresh agri-food products sold in traditional markets are still often marketed as unbranded goods, with no added value from innovations that can make these products different or better than competing products [38].

Currently, the business conditions of the agri-food industry in Central Europe are deteriorating due to the social restrictions imposed by the COVID-19 pandemic, which led to changes in the lifestyle and behavior of consumers who bought fresh produce from traditional markets during the pandemic. This has clearly had a negative impact on the supply chains of agri-food products in Central Europe [39]. The demand pattern for fresh produce is also disrupted, so that fresh produce that is not quickly absorbed by the market is at risk of deteriorating in quality [40,41]. Not only that, but the pattern of service experiences such as support, marketing and advice that agri-food product retailers normally provide to the end consumer is also affected [37,42].

This leads to a domino effect where fresh product performance and service performance are no longer optimally aligned with customer needs, negatively impacting customer satisfaction [43,44]. This must be taken into account in agri-food SMEs by implementing an open innovation strategy in order to obtain the most accurate information on what actions companies need to take in order to be able to innovate in satisfying their customers, customer satisfaction being a decisive factor in customers' purchasing decisions, which are a source of revenue for the survival of the enterprise.

The food industry is generally considered to have slow growth, low R&D investment, and a tendency to be conservative in innovation. Most consumers in the food industry are highly sensitive to products and consumption patterns. Add to this the stringent regulatory requirements for safety, and innovation in the food industry is a complex endeavor that requires a lot of time and effort. However, the dynamics of customer needs and the increasing number of new players, which further intensify competition, make innovation essential for any business and play an important role in the growth of the entire agricultural industry [45]. With so many stakeholders involved in innovation in the food industry, innovation needs to be carefully developed. An open innovation strategy needs to be implemented on a significant scale to access relevant information from external parties involved in the creation of knowledge and skills related to innovation in the food industry [46].

Yun et al. (2020) [47] explained that there are two main steps in implementing an open innovation strategy in the food industry. First, through communication by identifying information about customers and market needs. Detailed information is used to improve the quality of products (goods) and the creation of services. The second step is a more advanced step, namely the open innovation platform, which can integrate 'combinative innovation' and customer self-creation. This platform is designed to encourage combinations that can generate innovation.

### 2.2. SME Business Quality

SMEs need to understand that generating business continuity is a complex process and they need to be smart to find solutions in the form of strategies that will satisfy their customers and enable them to survive in the industry. This is because SMEs are not only competing with other SMEs but also with large companies in the same industry. This poses a great challenge to SMEs because the resources of SMEs are inherently smaller compared to those of large companies [48,49].

To compete with limited resources, SMEs need to define clear business objectives by finding out what their customers need from the enterprise. Using this information, SMEs can prioritize the allocation of their resources for product development or improvement and the enterprise's business processes according to customer expectations. Allocation of appropriate resources makes the business processes more efficient and the resulting products can be easily sold in the market as the specifications are based on the customer requirements. This can accelerate the growth of the SME business and reduce the risk of business failure [50].

The strategy of producing products and services with characteristics capable of meeting customer requirements is the application of quality in business or business quality. It begins with identifying customer requirements and then uses the information about customer requirements as the main consideration in developing or improving products, services, or business actions that continue throughout the business process to meet customer requirements [34,51]. Reyes-Menendez et al. (2019) [52] stated that the customer's perception of how the company responds to customer needs has a great impact on customer satisfaction and trust. The satisfaction customers feel will encourage them to make repeat purchases, then repeated purchases, always satisfied, will encourage customer satisfaction to develop into loyalty [53,54]. This is consistent with what was conveyed by Reyes-Menendez et al. (2018) [55], that the insight entrepreneurs have about what priorities customers want is an important basic asset to satisfy customers, which positively affects customer loyalty.

Therefore, enterprise quality can be one of the competitive factors for SMEs to differentiate themselves from competitors in order to survive. This is because business quality has a direct impact on customer decision-making and how customers perceive the enterprise based on their satisfaction [56,57].

### 2.3. Quality Function Deployment (QFD)

One of the most well-known methods that can help organizations improve quality is the quality function deployment (QFD) method. QFD is a method that establishes a continuous flow of information that ensures companies understand customer expectations for the product and service development process by integrating all of the enterprise's production systems that meet customer requirements. QFD is considered a very effective method to link the specifications of the product or service to be manufactured with the market needs. QFD can also be used to improve business integration between companies or between companies and their markets [58,59]. This was confirmed by Mukherjee (2019) [60], who stated that QFD is a popular approach to quality improvement. It can be a cross-functional/cross-departmental planning tool that applies customer requirements to all areas of an organization's business activities with the goal of maximizing customer satisfaction.

In its application, QFD uses tools in the form of a house of quality (HOQ) matrix that links the needs expressed by customers to any attributes and characteristics of a product in the form of goods or services by highlighting the existence of a relationship between the two elements [61]. Mostly QFD has been applied in the same or similar research areas, such as supply chain, product development and quality improvement [62]. However, several studies have also shown that QFD can be applied in other areas, such as sociology focusing on improvements in variables that can trigger marital loyalty [63], and public policy focusing on improving public services [64].

We adapted the main steps in QFD to achieve the goals of this study. We adapted the main steps into five main steps. Our first step is to determine customer requirements by identifying variables using an affinity diagram capable of collecting large amounts of verbal data, including ideas, opinions and problems, organized into groups based on the existence of an underlying relationship between them [65]. In our second step, we determine the priority and satisfaction level of customer requirements (CReqs). After determining the priority and satisfaction level of customer requirements, the third step is to use a tree diagram to translate the customer requirements (CReqs) into business requirements (BReqs). Tree diagrams are able to map different effective ways to solve customer requirement problems in order to achieve the main objectives related to quality [66]. Then, in the fourth step, the relationship between customer requirements (CReqs) and business requirements (BReqs) is determined using an L-shaped matrix that is capable of organizing a large number of features, functions and tasks of two parties in such a way that it can graphically represent logical connection points between CReqs and BReqs [67]. Certain weighted symbols are used to determine the weight of the relationship between customer requirements (CReqs) and business requirements (BReqs). The final step is then to evaluate the business quality to determine the priority level of business requirements (BReqs) and improvement actions based on the previously performed weighting of the business requirement (BReqs) variables, as shown in the Figure 1.

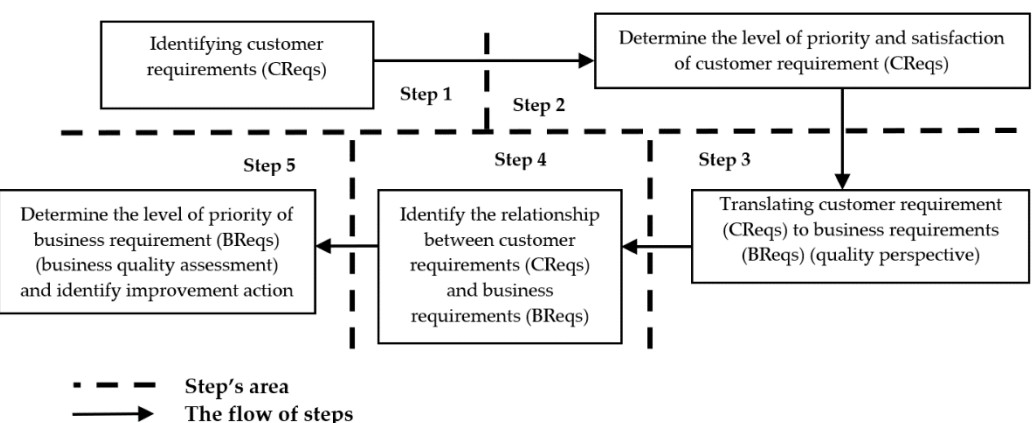

**Figure 1.** Research framework.

### 2.4. Research Framework

In various pieces of literature, it is clear that the application of business quality by translating customer requirements into business requirements so that businesses are able to produce goods and services that can meet customer requirements has an impact on customer satisfaction so that products can be easily absorbed into the market, and have an impact on the sustainability of the business itself to survive in the industry [68]. With regard to the implementation of quality in the SMEs, we divided this research framework into five main steps. The first step was to identify the customer requirements (CReqs), after identifying the CReqs variable, the second step was to determine the priority level and satisfaction level of the CReqs. The third step was to translate the CReqs into a quality perspective. The idea in this third step is to determine variables that are able to represent

the enterprise's perception of meeting customer needs, which are referred to as business requirements (BReqs) in this study. The fourth step was to determine how strong the relationship was between CReqs and BReqs. The final step was to perform an assessment to determine the priority of BReqs based on the relationship between CReqs and BReqs and then identify the relevant improvement actions.

The following research questions were compiled:

1.　What is the priority level of the customer requirement that needs to be improved?
2.　How is the priority level of business requirements to be improved based on the priority level of improving customer requirements?
3.　What are the business quality improvement actions based on the priority level of business requirements?

## 3. Methodology

### 3.1. Subjects and Measurement

This research was conducted in several traditional markets in Budapest, Hungary, the first being Great Market Hall Budapest, known as Central Market Hall Budapest (Nagy Vásárcsarnok). This market is also the center of tourism, so the consumers of this market are not only from Hungary, but also from different countries in the world. Second was Fehervar Road Fair Hall (Fehervar Uti Vásárcsarnok). Third was Bosnyak Ter Market Hall (Bosnyak Teri Vasarcsarnok), and fourth, Rakoczi Square Market (Rakoczi Teri Vásárcsarnok). Hungary is one of the countries that are able to make good use of EU funding for SME development. This is proven by the success of Hungary through the crisis that hit in 2008 [69,70].

The data collection lasted 4 months, starting in early December 2020 and ending in early April 2021. This study was a survey-type empirical study using a mix-method analysis that used primary data in the form of semi-structured in-depth interviews with five loyal customers and focus group discussions with three loyal customers who have purchased fresh agri-food products more than five times to identify the customer requirement variables and attributes (CReqs) used in the questionnaire items [71,72]. Then, these customer requirement variables (CReqs) and attributes were used as questionnaire survey items with a Likert scale to collect quantitative data from 268 customers and determine the degree of importance and satisfaction of the customer requirement variables (CReqs) (Table 1). Following Hair Jr. et al. (2014) [73], the minimum number of questionnaire survey samples in a study is 100 respondents. The number of samples in this study met the requirements.

**Table 1.** Questionnaire spread number based on area.

| Budapest | | | |
|---|---|---|---|
| Central Market Hall Budapest (Nagy Vásárcsarnok) | Fehervar Road Fair Hall (Fehervar Uti Vásárcsarnok) | Bosnyak Ter Market Hall (Bosnyak Teri Vasarcsarnok) | Rakoczi Square Market (Rakoczi Teri Vásárcsarnok) |
| 155 | 43 | 39 | 31 |

Figure 2 shows that the majority of customers selected as respondents were women (78%), while male respondents represented only 22%. In terms of age, the respondents were mainly in the age range of 41–50 years (39%); 31% of the respondents had an age range of 51–60 years, 16% had an age range of 31–40 years, 13% were in the age range of 20–30 years and 1% had an age range above 60 years. Among the respondents of the study, Hungarian citizens dominated (74%), while the remaining 26% were non-Hungarian citizens working in Hungary and some were also students studying in Hungary. In terms of the educational level of the respondents, middle school graduates dominated (78%); 14% were bachelor graduates and the other 8% were master graduates. Hungary is one of the developed countries that has a good and free education system. Hungarian government regulations in the field of education are able to encourage its citizens to pursue higher

education. The customers who were selected as respondents for the interview, focus group discussion and questionnaire survey were selected by the purposive sampling method. In this method, the respondent is a particular type of person who can provide the desired information because only that person has that information or because that person meets some of the criteria set by the researcher [74]. For the sample of this study, individuals were selected who shopped in Budapest from SMEs of fresh agricultural products.

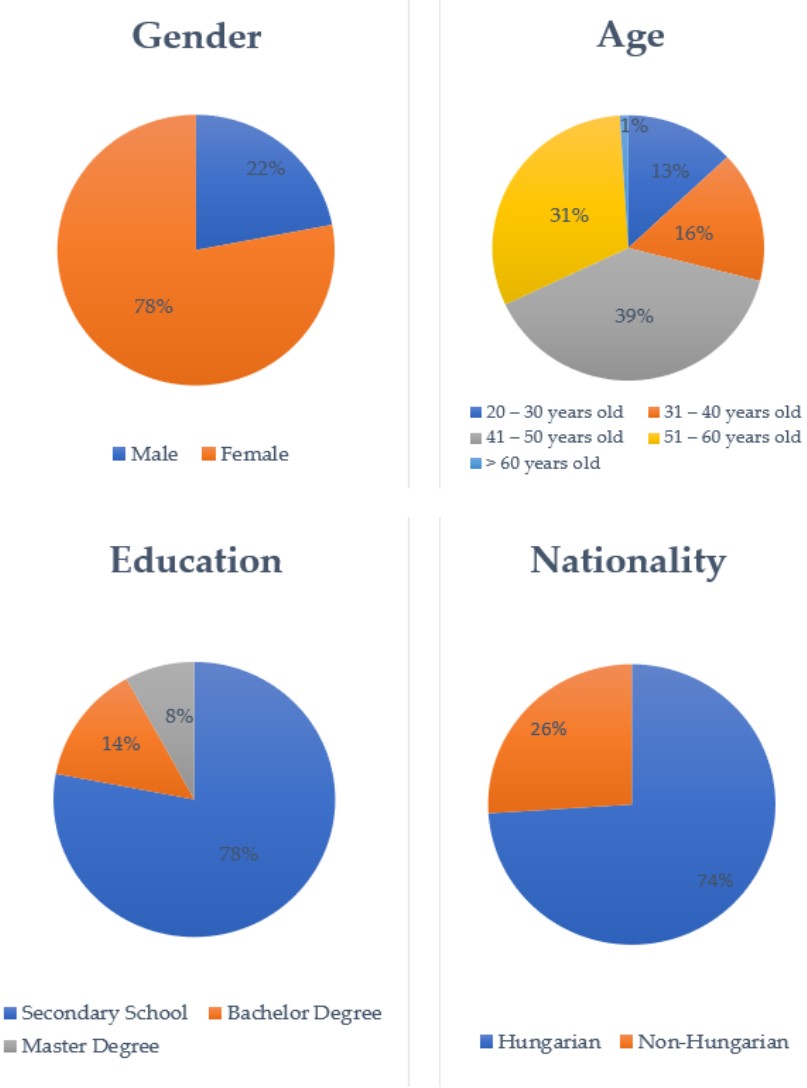

**Figure 2.** Number of questionnaire respondents based on gender, age, education and nationality.

The questionnaire survey used in this study was divided into three sections. The first section contained demographic characteristics of customers such as gender, age, education and nationality (see Figure 2). The second section contained 16 questions to measure level of importance of the customer requirement on a Likert scale of 1 = very unimportant and 5 = very important. The third section contained 16 questions to measure the level of satisfaction of customer requirement using a Likert scale 1 = very dissatisfied and 5 = very satisfied. The questionnaire was reviewed by academic experts.

In conducting this research, researchers always followed the health protocol rules required by the authorities as a guideline to avoid COVID-19, such as cleaning hands, using masks, applying social distancing, etc. Data processing for measuring reliability and validity was done using SPSS 25 (IBM, New York, NJ, USA). For the reliability test, the Cronbach alpha test was used, which is a general measure reliability test. The results of

the Cronbach alpha test showed a value of 0.847 for the questionnaire items at the level of importance and 0.807 for the questionnaire items at the level of satisfaction, which means that the data used were reliable. The test of data validity shows that each item is valid with a significance value of 0.01.

### 3.2. QFD Application

Heizer et al. (2017) [75] stated the implementation of QFD refers to determining which business actions are capable of satisfying customers and then translating customer needs into business process design. This process focuses on gathering rich information about customer needs to find new solutions for the business. The information gathered is integrated into the design of the company's business processes to determine where quality efforts should begin.

The QFD method uses the house of quality (HoQ), which is a tool in the form of a graphical technique to accurately translate the relationship between customer needs/requirements so that companies can design products, services and business processes to meet customer needs (Figure 3).

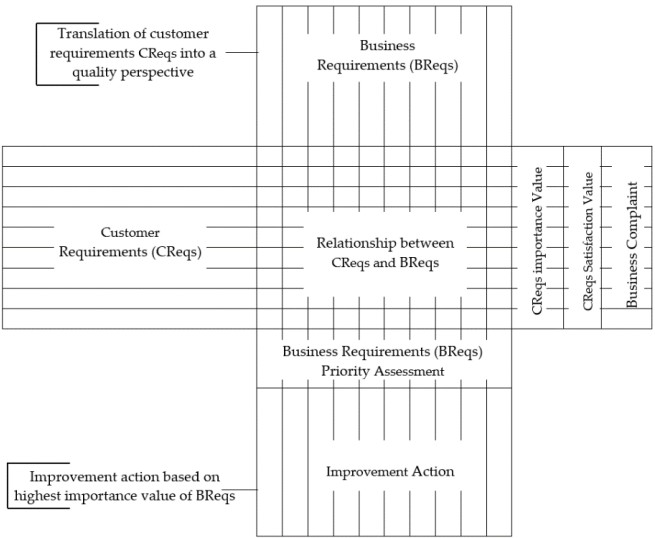

**Figure 3.** House of quality structure of business quality prioritization.

In recent years, the QFD method has become a reliable method in quality improvement research, such as the development of the QFD model proposed by Avikal et al. (2018) [31], which is based on the attributive classification of customer needs based on esthetic perception, enabling product improvement to increase customer satisfaction. The QFD method has also proven effective to be applied to quality improvement in the service industry, as demonstrated by the model proposed by Park et al. (2021) [27], which enables QFD to understand and reconcile the different perspectives between customers and service providers, which promotes improvements to meet customer expectations. This is further strengthened by the QFD model proposed by Pandey (2020) [29], that QFD can be a method to design business parameters to meet customer expectations. Therefore, the QFD method was used in this research.

## 4. Results and Discussion

### 4.1. Customer Requirements (CReqs)

Affinity diagrams are used extensively to analyze qualitative data from in-depth interviews and focus group among pilot study customers [65]. Customer requirements (CReqs) were identified as primary needs in this study. Further explains in detail which dimensions are included as CReqs. This dimension is called secondary needs. This dimension is called secondary needs. There are eight dimensions included in CReqs: price,

product performance, service, supply, recreation, hygiene and safety. Then, further analysis is done to identify variables that are able to better represent and explain each customer requirement (CReq) in more detail. This variable is called tertiary needs (Table 2).

**Table 2.** Customer requirements (CReqs) matrix.

| | | | Importance Value | Satisfaction Value | |
|---|---|---|---|---|---|
| Customer Requirement (CReqs) | Price | Suitable with ability to pay | 4.86 | 4.80 | |
| | | Competitive price | 4.83 | 4.73 | |
| | Product Performance | Freshness | 4.94 | 2.97 | Business Complaint |
| | | Texture | 4.85 | 3.00 | |
| | | Aroma | 4.80 | 3.10 | |
| | | Taste | 4.75 | 3.58 | |
| | Service | Pleasant interaction | 4.51 | 3.86 | |
| | Supply | Fast transaction | 4.39 | 4.02 | |
| | | Product availability | 4.80 | 4.17 | |
| | | Product variation | 4.62 | 4.48 | |
| | Recreation | Activity of leisure | 4.18 | 4.47 | |
| | | Activity of amusement | 4.17 | 4.16 | |
| | Hygiene | Product hygiene | 4.79 | 3.55 | Business Complaint |
| | | Market hygiene | 4.91 | 3.43 | |
| | Safety | Fraud-free transaction | 4.36 | 4.13 | |
| | | Crime-free shopping | 4.49 | 4.23 | |

The data we obtained from customers for the degree of importance and the degree of satisfaction were still in the form of large amounts of raw data. Each customer had a different perception of the degree of importance and the degree of satisfaction of each variable. So, we calculated the mean value of each variable. The simple formula we set up to calculate this generalized CReqs importance value is shown in the following equation:

$$\overline{CRIV}_i = \frac{\sum_{z=1}^{k} CRIV_z}{k} = \frac{CRIV_1 + CRIV_2 + CRIV_3 \ldots + CRIV_k}{k} \tag{1}$$

$\overline{CRIV}_i$ = customer requirements (CReqs) importance value, where *i* varies from 1 to *q*, where *q* is the number of CReqs tertiary requirements, which are eight variables.

$CRIV_z$ = customer requirements (CReqs) importance value collected from *z* customers as respondents, where *z* varies from 1 to *k*, where *k* is the number of customers as respondents (268 people).

$$\text{Suitable with ability to pay's importance value} = \frac{\sum_{z=1}^{k} CRSV_z}{k} = \frac{1302}{268} = 4.86 \tag{2}$$

$$\text{Competitive price's importance value} = \frac{\sum_{z=1}^{k} CRSV_z}{k} = \frac{1295}{268} = 4.83 \tag{3}$$

$$\text{Freshness's importance value} = \frac{\sum_{z=1}^{k} CRSV_z}{k} = \frac{1324}{268} = 4.94 \tag{4}$$

$$\text{Texture's importance value} = \frac{\sum_{z=1}^{k} CRSV_z}{k} = \frac{1300}{268} = 4.85 \tag{5}$$

$$\text{Aroma's importance value} = \frac{\sum_{z=1}^{k} CRSV_z}{k} = \frac{1287}{268} = 4.80 \tag{6}$$

$$\text{Taste's importance value} = \frac{\sum_{z=1}^{k} CRSV_z}{k} = \frac{1273}{268} = 4.70 \tag{7}$$

$$\text{Pleasant interaction's importance value} = \frac{\sum_{z=1}^{k} CRSV_z}{k} = \frac{1209}{268} = 4.51 \tag{8}$$

$$\text{Fast transaction's importance value} = \frac{\sum_{z=1}^{k} CRSV_z}{k} = \frac{1174}{268} = 4.39 \tag{9}$$

$$\text{Product availability's importance value} = \frac{\sum_{z=1}^{k} CRSV_z}{k} = \frac{1287}{268} = 4.80 \tag{10}$$

$$\text{Product variation's importance value} = \frac{\sum_{z=1}^{k} CRSV_z}{k} = \frac{1239}{268} = 4.62 \tag{11}$$

$$\text{Activity of leisure's importance value} = \frac{\sum_{z=1}^{k} CRSV_z}{k} = \frac{1121}{268} = 4.18 \tag{12}$$

$$\text{Activity of amusement's importance value} = \frac{\sum_{z=1}^{k} CRSV_z}{k} = \frac{1118}{268} = 4.17 \tag{13}$$

$$\text{Product hygiene's importance value} = \frac{\sum_{z=1}^{k} CRSV_z}{k} = \frac{1284}{268} = 4.79 \tag{14}$$

$$\text{Market hygiene's importance value} = \frac{\sum_{z=1}^{k} CRSV_z}{k} = \frac{1316}{268} = 4.91 \tag{15}$$

$$\text{Fraud} - \text{free transaction's importance value} = \frac{\sum_{z=1}^{k} CRSV_z}{k} = \frac{1169}{268} = 4.36 \tag{16}$$

$$\text{Crime} - \text{free shopping's importance value} = \frac{\sum_{z=1}^{k} CRSV_z}{k} = \frac{1204}{268} = 4.49 \tag{17}$$

CReqs satisfaction value is provided in the following equation:

$$\overline{CRSV_i} = \frac{\sum_{z=1}^{k} CRSV_z}{k} = \frac{CRSV_1 + CRSV_2 + CRSV_3 \ldots \ldots + CRSV_k}{k} \tag{18}$$

$\overline{CRSV_i}$ = customer requirements (CReqs) satisfaction value, where $i$ varies from 1 to $q$, where $q$ is the CReqs tertiary needs number, which is eight variables.

$CRSV_z$ = customer requirements (CReqs) satisfaction value collected from $z$ customers as respondents, where $z$ varies from 1 to $k$, where $k$ is the number of customers as respondents (268 people).

$$\text{Suitable with ability to pay's satisfaction value} = \frac{\sum_{z=1}^{k} CRSV_z}{k} = \frac{1288}{268} = 4.80 \tag{19}$$

$$\text{Competitive price's satisfaction value} = \frac{\sum_{z=1}^{k} CRSV_z}{k} = \frac{1268}{268} = 4.73 \tag{20}$$

$$\text{Freshness's satisfaction value} = \frac{\sum_{z=1}^{k} CRSV_z}{k} = \frac{607}{268} = 2.97 \tag{21}$$

$$\text{Texture's satisfaction value} = \frac{\sum_{z=1}^{k} CRSV_z}{k} = \frac{805}{268} = 3.00 \tag{22}$$

$$\text{Aroma's satisfaction value} = \frac{\sum_{z=1}^{k} CRSV_z}{k} = \frac{833}{268} = 3.10 \tag{23}$$

$$\text{Taste's satisfaction value} = \frac{\sum_{z=1}^{k} CRSV_z}{k} = \frac{959}{268} = 3.58 \tag{24}$$

$$\text{Pleasant interaction's satisfaction value} = \frac{\sum_{z=1}^{k} CRSV_z}{k} = \frac{1035}{268} = 3.86 \tag{25}$$

$$\text{Fast transaction's satisfaction value} = \frac{\sum_{z=1}^{k} CRSV_z}{k} = \frac{1078}{268} = 4.02 \tag{26}$$

$$\text{Product availability's satisfaction value} = \frac{\sum_{z=1}^{k} CRSV_z}{k} = \frac{1118}{268} = 4.17 \tag{27}$$

$$\text{Product variation's satisfaction value} = \frac{\sum_{z=1}^{k} CRSV_z}{k} = \frac{1201}{268} = 4.48 \qquad (28)$$

$$\text{Activity of leisure's satisfaction value} = \frac{\sum_{z=1}^{k} CRSV_z}{k} = \frac{1198}{268} = 4.47 \qquad (29)$$

$$\text{Activity of amusement's satisfaction value} = \frac{\sum_{z=1}^{k} CRSV_z}{k} = \frac{1115}{268} = 4.16 \qquad (30)$$

$$\text{Product hygiene satisfaction's value} = \frac{\sum_{z=1}^{k} CRSV_z}{k} = \frac{952}{268} = 3.55 \qquad (31)$$

$$\text{Market hygiene's satisfaction value} = \frac{\sum_{z=1}^{k} CRSV_z}{k} = \frac{920}{268} = 3.43 \qquad (32)$$

$$\text{Fraud} - \text{free transaction's satisfaction value} = \frac{\sum_{z=1}^{k} CRSV_z}{k} = \frac{1107}{268} = 4.13 \qquad (33)$$

$$\text{Crime} - \text{free shopping's satisfaction value} = \frac{\sum_{z=1}^{k} CRSV_z}{k} = \frac{1134}{268} = 4.23 \qquad (34)$$

The improvement actions proposed in this study are based on priority CReqs variables that are considered important by consumers, in that they have an importance value above 4, but at the same time they are not able to satisfy customers marked with a value below 4, which we call a "business complaint", as shown in Table 2.

Price describes the price-related considerations of the customer when selecting goods or services for purchase. Price has a significant impact on perceived utility by comparing the value of the product with the price to be paid. Affordable prices make customers more satisfied and loyal to the enterprise [76,77] and comprise two tertiary needs which are suitable with the ability to pay and competitive price. The tertiary needs of price are included in the top five variables with the highest importance value.

Next, product performance had the most tertiary needs: product freshness, texture, aroma and taste. Product performance refers more to the characteristics of the food product in meeting customer satisfaction and needs [78]. The respondents' results show that the tertiary need for product performance, namely freshness, is considered the most important but has the lowest satisfaction value. The other three tertiary needs included in the six lowest satisfaction variables were texture, aroma and taste, which were not considered to be able to satisfy customers. This was indicated by the values below 4 which were entered into the "business complaint" matrix.

The next secondary need is service, which deals with the ability of SME food services to satisfy customers [79]. Service insight value on food SMEs comes from interactions between sellers and buyers that produce service values that customers need to be satisfied. In this research, it had two tertiary needs, pleasant interaction and fast transaction [50]. The results show that pleasant interaction was not satisfactory, based on a satisfaction score of less than 4. Meanwhile, fast transaction had a value above 4, indicating that it had satisfactory performance.

Our other secondary need, which is recreation, is more about customer requirements to make shopping a fun activity that fills leisure time [80]. Here there are two tertiary needs, which are activity of leisure and activity of amusement. Neither of the tertiary needs had a value below 4, which means that customers were satisfied with the recreational activities they experience while shopping.

The supply variable is more concerned with the availability of agricultural and food products. This refers to the availability of products that are needed by customers and the availability of variations of fresh agricultural and food products that are desired by customers. The availability of a food product must be based on changes in customer interest in food trends [47]. Supply has two tertiary needs, which are product availability and product variation. The results show that product availability and product variation are essential tertiary needs for customers and have satisfactory performance.

When it comes to hygiene, there are two tertiary needs, which are product hygiene, having the second highest importance score, and market hygiene, which tends to address the desire for product cleanliness and the experience of shopping in a clean place [81]. Both tertiary needs had a satisfaction score below 4.

The last secondary need is safety with two tertiary needs, namely fraud-free transactions and crime-free shopping, relating to customer requirements in terms of feeling secure when shopping offline. The results of the respondents show that the values of the two tertiary needs were not less than 4, which means that they had a good safety performance.

### 4.2. Business Requirements (CReqs)

Tree diagrams were used to translate customer requirements (CReqs) into business requirements (BReqs). Tertiary needs should be translated into the types of technical language, terms or relevance of quality used in the system, organization and management. Business requirements (BReqs) represent solutions that relate to improvement actions for the organization to take [82].

As shown in Table 3, the translation matrix from CReqs to BReqs shows that the tertiary requirements of customer requirements (CReqs) include suitable with the ability to pay and competitive price, which are relevant to affordability, then freshness, which is relevant to shipping time, and texture, which is relevant to storage, then aroma and taste, relevant to product grade, and pleasant interactions, which is relevant to attitude and so on.

**Table 3.** Customer requirement (CReqs) to business requirements (BReqs) translation matrix.

| No. | Tertiary Needs | Business Requirement (BReqs) |
|-----|----------------|------------------------------|
| 1 | Suitable with ability to pay | Affordability |
| 2 | Competitive price | |
| 3 | Freshness | Shipping time |
| 4 | Texture | Storage |
| 5 | Aroma | Product grade |
| 6 | Taste | |
| 7 | Pleasant interactions | Attitude |
| 8 | Fast transaction | Responsiveness |
| 9 | Product availability | Distribution |
| 10 | Product variation | Network |
| 11 | Activity of leisure | Decoration |
| 12 | Activity of amusement | Conversation |
| 13 | Product hygiene | Sanitary |
| 14 | Market hygiene | |
| 15 | Fraud-free transaction | Credibility |
| 16 | Crime-free shopping | Security |

### 4.3. Business Quality Assessment

Each relationship between CReqs and BReqs is defined with quantified relationship symbols to facilitate readability. This study uses three scale symbols, "strong", "moderate" and "weak". Symbols with a weight of 9 are used for strong relationships, symbols with a weight of 3 are used for moderate relationships, and symbols with a weight of 1 are used for weak relationships (Table 4).

**Table 4.** Relationship scale symbol.

| Level of Relationship | Graphic Symbol | Value |
|:---:|:---:|:---:|
| Strong | ● | 9 |
| Moderate | ○ | 3 |
| Weak | △ | 1 |

A strong relationship indicates that there is a relationship between the CReqs variable and certain BReqs that share the same characteristics and are directly involved. A moderate relationship indicates that the variables have almost the same characteristics and are directly involved. A weak relationship indicates that the variables have characteristic similarities and have indirect involvement that supports certain variables.

To determine the priority level of the BReqs, an assessment was performed by summing the multiplication value of the CReqs importance value with the quantified relationship value between the CReqs and the BReqs variable in the business complaint area indicated by the relationship scale symbol (Figure 4). The simple formula we created to calculate the BReqs importance value is shown in the following equation:

$$BRV_i = \sum_{i=1}^{n} \overline{CRIV}_{iBusComp} \, QR_j \qquad (35)$$

$BRV_i$ = business requirements (BReqs) value where $i$ varies from 1 to $q$, where $q$ is the number of BReqs variable.

$\overline{CRIV}_{iBusComp}$ = customer requirements (CReqs) importance value collected from *iBusComp* customers importance value, where *iBusComp* varies from 1 to $n$, where $n$ is the number of customer requirements importance value in the business complaint area of the house of quality matrix (Figure 4).

$QR_j$ = quantified relationship between $\overline{CRIV}_{iBusComp}$ and business requirements (BReqs) variables where $j$ varies from 1 to $m$, where $m$ is the number of business requirements (BReqs) variable. $QR_j$ value is indicated by symbols as shown in Table 4.

$$\text{Product grade's value} = ((3(4.94) + (3(4.85) + (9(4.80) + (9(4.75)) = 115.3 \qquad (36)$$

$$\text{Shipping time's value} = ((9(4.94) + (3(4.85) + (3(4.80) + (3(4.75)) = 87.7 \qquad (37)$$

$$\text{Sanitary's value} = ((9(4.79) + (9(4.91)) = 87.3 \qquad (38)$$

$$\text{Storage's value} = ((3(4.94) + (9(4.85) + (3(4.80) + (3(4.75)) = 87.1 \qquad (39)$$

$$\text{Attitude's value} = (9(4.51)) = 40.6 \qquad (40)$$

The results show that this study succeeded in identifying CReqs from a quality perspective. Moreover, adopting measures from QFD not only results in a perspective, but also prioritizes some improvement actions among the variables. These measures are simply represented with symbols to make them easier to understand as shown in Figure 4.

Related to the first research question, there are three dimensions that need to be improved, namely "product performance", "service" and "hygiene". The first dimension shows that all variables need to be improved, which are freshness, texture, aroma and taste. The second dimension shows that there is one variable that needs to be improved, which is pleasant interaction. The third dimension shows that there are two variables that need to be improved, which are product hygiene and market hygiene.

Related to the second research question, there are five priority perspectives of business requirements to be improved. Only the values of the variables BReqs, which have the same characteristics indicated by a strong relationship symbol associated with the priority dimensions and variables of CReqs, are used. These are marked with the gray area associated with the business complaint matrix. They include product grade as the top

priority, shipping time as the second priority, sanitary as the third priority, storage as the fourth priority and attitude with the lowest value as the fifth priority.

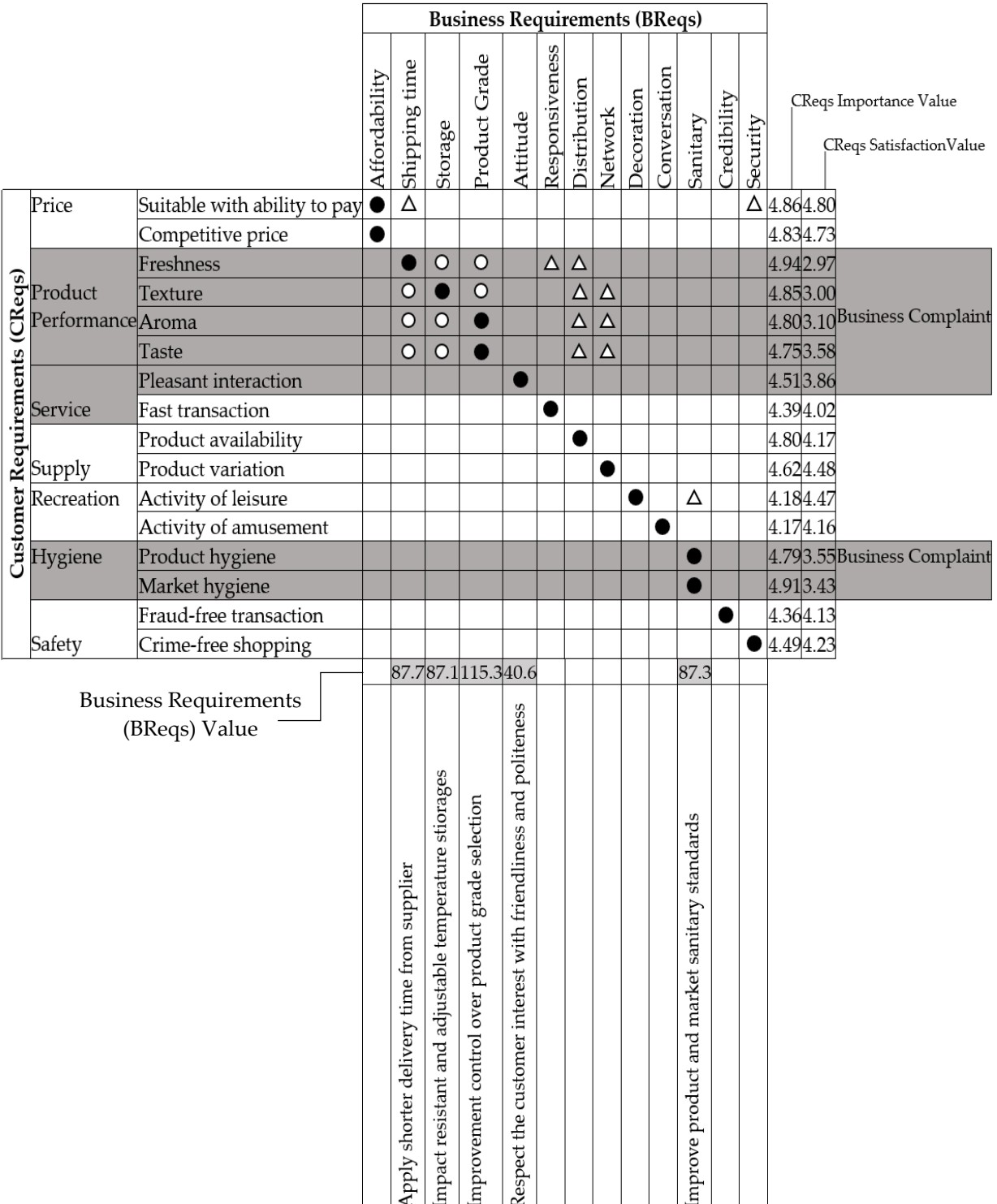

**Figure 4.** The house of quality matrix of business quality improvement priority (final).

Related to the third research question, we used a shipping time perspective to improve the freshness variable. The recommended action is to emphasize shortening the

shipping time of products from the supplier to the retail market in order to maintain product freshness.

From the perspective of storage, this study attempts to offer measures to improve texture by selecting or changing storage locations that can protect agri-food products from harsh influences that may occur in the distribution process and from extreme temperatures.

The action measure of aroma and flavor is to improve the control of class selection in the product. Retailers need to control the products supplied by suppliers by improving the previous standard class of products, by selecting with new standards without increasing the prices.

Pleasant interaction has relevance to attitude: the proposal is SMEs must be able to respect the interests of customers and serve customers pleasantly and courteously. A good attitude can leave a positive impression on customers, which provides comfort in shopping.

The last two variables are product hygiene and market hygiene, which determine customer convenience in shopping based on customer requirements for cleanliness. In this study, these two variables are considered from a sanitary perspective. SMEs need to improve sanitary controls on each product supplied and clean it before sale. SMEs also need to work with market managers to improve hygiene standards in markets by increasing the schedule and staffing for market cleaning and providing more hand sanitizer on-site to increase customers' confidence that they can safely shop in traditional markets even during a pandemic situation.

In keeping with this, Yost and Cheng (2021) [83] and Goswami and Chouhan (2021) [84] confirm that the impact of COVID-19 affects the perception of food industry customers by placing food security, quality and safety, in this case food availability, nutrition and sanitation, as primary requirements for food vendors. Food suppliers/vendors need to build trust by maintaining good relationships with suppliers to ensure their product availability and providing a clean and safe place protected from infection hazards. Increased customer trust is able to increase the survivability of SMEs as they have the ability to attract customers.

On the other hand, the results of Hobbs (2021) [85] also confirm that the disruption caused by the impact of the pandemic COVID-19 causes an imbalance in the intermediate market that is closely related to the resilience of the supply chain, so that the decision on actions related to the agri-food supply chain strongly affects product quality, availability of agri-food products and sales prices during the pandemic. The agri-food industry experienced a significant downturn in the last quarter of 2020, affecting the livelihoods of millions of people in the agri-food industry worldwide. This pandemic is emanating from the agri-food market, hence real improvement measures are required that are capable of satisfying the customers to restore the confidence of the customers in the agri-food market [86].

Finally, the results of this study are able to present real solutions in the form of business action priorities as a whole and in detail that can be taken by agri-food SMEs to meet the priority needs of agri-food customers. The surprising thing is that it turns out that price-related variables are not the main concern for customers during the pandemic. This shows that customers are willing to pay for products at a reasonable price as long as their needs for product performance, service and hygiene can be met. In addition, this research is also able to reveal customer needs for recreation that are considered important during a pandemic. The lockdown policy that forces people to continue to stay at home certainly increases the boredom of customers who need activities of leisure and entertainment. Not only that, but the lockdowns and social restrictions imposed by local governments are having an impact on data collection limitations, as a number of markets experienced a drastic drop in visitor numbers. This research is able to open opportunities for other researchers to conduct more focused quality research, such as specific research to improve the quality of agricultural products and fresh food, especially in the application of the latest technology, and supply chain resilience, which can provide further solutions to the issues among agri-food SMEs.

### 4.4. Agri-Food SMEs Business Quality Improvement through Open Innovation

Business quality improvement requires information about customer requirements, which is needed by SMEs to be able to innovate business quality improvement. Spithoven et al. (2013) [87] noted that adopting inbound open innovation principles in delivering quality is important for SMEs. Exploration of external sources of information in this research is that information on customer needs can be used by SMEs to determine business actions and innovate to improve the quality of their business effectively and efficiently, as SMEs still struggle with the problem of limited resources. This has also been confirmed by Sarkar and Costa (2008) [45], who found that the application of the open innovation business model has proven to be applicable and enables increased innovation effectiveness and business efficiency in the food industry. Collaboration with customers in seeking information, building good relationships with suppliers and other stakeholders is necessary to realize the potential of open innovation.

In reality, however, the implementation of open innovation as a whole is still a real challenge for SMEs in the food industry, as they have little ability to mitigate the risks of existing challenges, little collaboration, insufficient human resources and limited access to financial resources. SMEs' perception that open innovation will be very expensive to implement overall leads them to use only open innovation strategies that focus on short-term profitability. SMEs need to change the mindset that assumes there is a new paradigm and approach that is capable of delivering change for both industry and SMEs. Company size offers the advantage of flexibility in building and maintaining relationships when working with external parties. Limited resources can be leveraged by SMEs to leverage key resources from partner companies to facilitate innovation in generating value to satisfy customers [88].

This study shows that open innovation in the food industry provides dynamic results, especially in terms of information about the level of satisfaction and importance of customer needs. The results show that a proportion of customers are dissatisfied with product performance, service and hygiene. This is quite surprising considering that this research was conducted in developed countries with advanced agri-food technology. This is related to the changing quality standards in customer perception, which are very dynamic in the pandemic situation. Customers will require high-quality agri-food products sufficient for nutritional needs to increase immunity, services that provide a sense of comfort without stress, and hygiene to avoid the possibility of disease infection when shopping during a pandemic. SMEs need to maintain a strong relationship with their customers by really knowing and understanding what the customers really want in order to be able to meet their customers' needs. This relationship plays an important role in generating open innovation [89].

The adaptation of business processes, resources and market conditions of each food industry may be different, so it will produce different open innovations. Yun et al. (2020) [47] gave a case study on the application of open innovation in restaurants, first on open food innovation conducted in a restaurant in Italy. The focus of open innovation is to innovate a variety of menus, ensure excellent service and amazing place decorations. Next, they studied a restaurant in South Korea, which implements open innovation by offering a variety of food and trying to establish good communication with customers to get the necessary information that can be brought into the enterprise's business processes to meet customer needs. Both restaurants have the same purpose, which is to create interactions that can generate information about customers and markets, and to foster communication with customers, suppliers and other employees to obtain accurate information that can be used by SMEs/business actors in the food industry to compete.

Yun (2017) [90] offers a dynamic model of open innovation, which in combination with the application of business quality is suitable to be used as the right business strategy for SMEs in anticipating the rapid dynamics and complexity of factors that may affect the business. Logic suggests adopting a more adaptable inbound open innovation initial concept that is able to analyze dynamic effects to develop and increase the competitive

advantage of the enterprise by being able to provide the set of product or service features that are able to meet customer needs effectively and efficiently. By integrating their resources, agri-food SMEs are able to innovate and create social safety nets for their business continuity [91].

## 5. Conclusions

This study result shows that there are five main priority actions on business requirements (BReqs) in sequence. First, improving control over product variety selection. Second, arranging shorter delivery times from suppliers. Third, improving product and market hygiene standards. Fourth, introducing impact resistant and adjustable temperature storage. Fifth, respecting customer interests with friendliness and politeness. These five measures are considered to be able to achieve improvements in the variables of freshness, texture, aroma, taste, pleasant interaction, product hygiene, and market hygiene, which are seven priorities for customer needs that were previously included in the business complaint variable because they were not considered to be able to satisfy customers.

Determining the priority of customer requirements and the priority of business requirements plays an important role for SMEs seeking to improve their business quality. By capturing the business quality conditions of SMEs, they are able to take the right and most important actions for their business so that the allocation of resources owned by SMEs can be improved in a targeted manner.

The process of identifying customer requirements (CReqs) carried out in this study demonstrates the important role of the principle of open innovation in opening up the flow of information that the agri-food SME market needs to develop innovations to meet customer needs. Appropriate resource allocation is important for SMEs given their limited resources and the high complexity of the characteristics of the fresh agri-food business. This helps SMEs deal with the unclear and confusing circumstances of running and managing a business at a time of deteriorating economic conditions during the current pandemic.

### 5.1. Implication

Several contributions have been made in this research. The results of this study are relevant to SMEs as they suggest entrepreneurial actions that can improve business quality. This research presents a new approach to SME agri-food businesses that focuses on quality and contributes to a wider acceptance of the QFD method. Second, the application of the QFD method in agri-food SMEs shows how implementing a quality strategy can help SMEs adapt to business conditions and determine business actions to survive in the industry. The implications of this research are divided into three parts, namely theoretical, practical and methodological implications as shown in the following summary Table 5.

### 5.2. Limitation

The social restriction schemes put in place by the local government at the time of this research have meant that this study has limitations in terms of data collection. A number of markets have seen a drastic drop in visitor numbers, so data collection will be more optimal under normal conditions.

### 5.3. Future Research

This research presents an analysis of business quality in general, so this provides an opportunity for other researchers to conduct more focused quality research, such as specific research to improve the quality of fresh agricultural and food products, particularly in the application of current technology, and supply chain resilience which can provide solutions for more detailed dimensions of product quality taking into account the results of this study. It shows that product quality is one of the main customer requirements that are considered important but have not been able to satisfy customers so far.

<div align="center">

**Table 5.** Summary table of research implications.
</div>

| Area | Implication |
|---|---|
| Theoretical Implication | • Variables related to price are not the main concern of customers, but product quality is the main concern of customers when shopping.<br>• Shipping time, storage, product grade, attitude and sanitary variables are the most important considerations related to quality perception in determining the actions to be taken by the enterprise to meet customer requirements. |
| Practical Implication | • SMEs can focus their resource allocation on improvements in shipping time, product quality, service attitude and sanitary conditions during a pandemic.<br>• The application of quality strategy analysis can help SMEs map business conditions and determine the best business actions to survive in the industry. |
| Methodology implication | • A new theoretical approach contributes to the wider application of the quality function deployment method (QFD). |

**Author Contributions:** Conceptualization, T.W.; methodology, T.W..; software, T.W.; validation, T.W., M.B.H. and C.B.I.; formal analysis, T.W.; investigation, T.W.; resources, T.W.; data curation, T.W.; writing—original draft preparation, T.W.; writing—review and editing, T.W.; visualization, T.W.; supervision, C.B.I.; project administration, T.W.; funding acquisition, T.W., M.B.H. and C.B.I. All authors have read and agreed to the published version of the manuscript.

**Funding:** This research received no external funding.

**Institutional Review Board Statement:** Not applicable.

**Informed Consent Statement:** Not applicable.

**Data Availability Statement:** The data presented in this study are available on request from the corresponding author.

**Acknowledgments:** First we would like to thank the reviewers who gave us suggestions for the further development of this article. Secondly, we would like to thank the customers of agri-food SMEs in Hungary who gave us the opportunity to collect data and agreed to be interviewed.

**Conflicts of Interest:** The authors declare no conflict of interest.

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
