# Peer review of "Prioritizing Business Quality Improvement of Fresh Agri-Food SMEs through Open Innovation to Survive the Pandemic: A QFD-Based Model"

_2199-8531, doi:10.3390/joitmc7020156_

Round 1
Reviewer 1 Report
The paper, “Prioritizing Business Quality Improvement of Fresh Agri-food 3 SMEs through Open Innovation to Survive the Pandemic: A 4 QFD-based Model”, addresses a research area interesting in agri-food SMEs based on quality perspective. However, some aspects could be considered about TQM in SMEs sector to reinforce the literature review.
The authors draw on several prior studies, but a much more critical literature analysis is needed to strengthen the paper’s argument and draw out the gaps they seek to address. Which gap(s) in extant studies are authors trying to address here? The authors should be present this gap also in Introduction section.
While the author(s) establish some links to some literature, author(s) need to establish a more coherent framework for the overall paper and more recent literature can be used.
The paper lacks some appropriate methodological procedures hence the text tends to wander and even approach waffle at certain points… More explanation is need about the sample selection, type of study, and data selection instrument as well as data a analysis.
Also, the paper needs to be present much stronger discussion and conclusion sections in order to offer value to the reader. There are a number of very findings that make it unwieldy for a smooth reading….
The conclusions and implications could be reinforced, innovative and more contributions for SME owner-manager and public policy is needed.
A typo:
Pag. , paragraph 4,there is two similar statements.
Author Response
Point 1: The paper, “Prioritizing Business Quality Improvement of Fresh Agri-food 3 SMEs through Open Innovation to Survive the Pandemic: A 4 QFD-based Model”, addresses a research area interesting in agri-food SMEs based on quality perspective. However, some aspects could be considered about TQM in SMEs sector to reinforce the literature review.
Response 1: Broadly speaking, TQM is a customer-focused quality management system with continuous improvement, we have explained its application to SMEs in the “2.2 SMEs Business Quality” section, lines 141-159.
Point 2: The authors draw on several prior studies, but a much more critical literature analysis is needed to strengthen the paper’s argument and draw out the gaps they seek to address. Which gap(s) in extant studies are authors trying to address here? The authors should be present this gap also in Introduction section.
Response 2: We have added more literature on introduction section (line 73-87). In its application, QFD has been successfully used by many researchers in various fields However, the QFD approach has never been used by research related to the application of business quality in the agri-food industry. This study aims to fill this gap and apply the QFD method to prioritize enterprise quality improvement in the mid-sized agri-food industry.
Point 3: While the author(s) establish some links to some literature, author(s) need to establish a more coherent framework for the overall paper and more recent literature can be used.
Response 3: We have added more on literature review (line 160-163, 165-167) and we added more explanation to establish more coherent framework (line 217-227)
Point 4: The paper lacks some appropriate methodological procedures hence the text tends to wander and even approach waffle at certain points… More explanation is need about the sample selection, type of study, and data selection instrument as well as data a analysis.
Response 4: This study is a survey-type empirical study using a mix-method analysis that uses primary data in the form of semi-structured in-depth interviews with 5 loyal customers and focus group discussions with 3 loyal customers who have purchased fresh agri-food products more than five times to identify the customer requirements variables and attributes (CReqs) used in the questionnaire items. Then, these customer requirement variables (CReqs) and attributes were used as questionnaire survey items with a Likert scale to collect quantitative data from 268 (line 261-270).
The customers who were selected as respondents for the interview, focus group discussion and questionnaire survey were selected by purposive sampling method (line 307-308),
Point 5: Also, the paper needs to be present much stronger discussion and conclusion sections in order to offer value to the reader. There are a number of very findings that make it unwieldy for a smooth reading….
Response 5: We have added More explanation and modification to make stronger discussion (line 571-598)
Point 6: The conclusions and implications could be reinforced, innovative and more contributions for SME owner-manager and public policy is needed.
Response 6: We have added more explanation on the implication (line 707-716)
Point 7: A typo: Pag. , paragraph 4,there is two similar statements.
Response 7: We have corrected paragraph 4, which previously occurred typo two similar statements (line 61-63)

Reviewer 2 Report
This research is very interesting and the work is well structured and developed.
The introduction contextualizes the research framework very well and the topics for the literature review section are very well selected. In addition, a large number of bibliographical references of interest are provided.
The presentation of the results is very clear, highlighting the contribution of the paper. However, in the conclusions section, 6.1., a summary table could be added with the recommendations or implications for companies, since it is the main contribution of the research and it would be convenient to highlight it.
Author Response
Point 1: The presentation of the results is very clear, highlighting the contribution of the paper. However, in the conclusions section, 6.1., a summary table could be added with the recommendations or implications for companies, since it is the main contribution of the research and it would be convenient to highlight it.
Response 1: based on your review and suggestions we have added a summary table as you wish (line 713-716).

Reviewer 3 Report
Thank you for giving me the possibility to review this paper. I hope that the authors find my comments productive and that they help them to improve their research work.
In this paper the authors carry out a study to identify the priority business requirements and the improvement and innovation actions that are needed to generate business quality in agri-food SMEs to satisfy customer needs.
The proposed Keywords are correct and correspond to the research topic, however the keyword agri-food should be included.
The authors need to be cited correctly in lines 26, 34, 105, 125, 125, 166, ... where the date is missing. All citations need to be checked.
The Introduction should explain all the keywords that will be used in the paper, for example "agri-food" and the authors should indicate the main motivations for carrying out this research. The objective of the paper is well explained in the penultimate paragraph of this section, but the research questions should be in the Literature review section (see comment below). The structure of the paper is correct in the last paragraph.
In the literature review, the authors are asked to review Yun's citation (line 125) as the second step that is mentioned does not appear. At the end of the Literature review section, the authors should include the research questions formulated in the Introduction, where they should detail the problem or gap that has been detected and on which a prediction has been made that later, in the methodology, has been corroborated or not. In addition, authors are asked to bear in mind that the research questions should be based on the literature review.
In Methodology the sample and data collection is well explained, however, authors are asked to reflect the data in Figure 2 in the text and to include this Figure after the paragraph where it is mentioned and not before.
The author should also justify the choice of questions in the questionnaire. Authors also have to include a table with statistics in the analysis of the results such as Reyes-Menendez, A., Palos-Sanchez, P. R., Saura, J. R., & Martin-Velicia, F. (2018). Understanding the influence of wireless communications and Wi-Fi access on customer loyalty: a behavioral model system. Wireless Communications and Mobile Computing, 2018.
In addition, the authors are asked to elaborate on the theoretical models on which this methodology has been based, such as those carried out by the authors Reyes-Menendez, A., Saura, J. R., & Martinez-Navalon, J. G. (2019). The impact of e-WOM on hotels management reputation: exploring tripadvisor review credibility with the ELM model. IEEE Access, 7, 68868-68877.
For example in point 3.2, the QFD method should be referenced, as should the statement in point 4.1 when it says "Affinity diagrams are used extensively (...)".
Although the results are correct, it is suggested that the author present them in a clearer way. For example, presenting on one side the research question, the result and whether or not it has been fulfilled. The wording of this section is sometimes a bit confusing.
Likewise, the justification of the formula used in this same point, 4.1, (line 342) or in line 431 of section 4.3. Regarding section 4.3, the data should be represented and explained beyond just showing the formula.
In the results, the author is also asked to include a brief summary of the main conclusions drawn from the results.
The results are correctly explained, but need the above-mentioned improvement.
In the Discussion section the authors should include limitations and future research (points 6.2 and 6.3).
The Conclusion section is intended to clarify the objectives of the research, to state the aim of the research and to show what the authors are demonstrating with their research. Authors are asked to develop this section accordingly.
The references are correct and most of them are up to date (from 2016 to the present).
Author Response
Point 1: The proposed Keywords are correct and correspond to the research topic, however the keyword agri-food should be included.
Response 1: We have added agri-food on the keyword
Point 2: The authors need to be cited correctly in lines 26, 34, 105, 125, 125, 166, ... where the date is missing. All citations need to be checked.
Response 2: citation checked and we have added date/year on the citation (moved to line 26, 33, 74, 75, 77, 79, 81, 112, 132, 160, 165, 269, 332, 343, 347, 663).
Point 3: The Introduction should explain all the keywords that will be used in the paper, for example "agri-food"
Response 3: Explanation about agri-food in the introduction (line 38-53)
Point 4: In the literature review, the authors are asked to review Yun's citation (line 125) as the second step that is mentioned does not appear.
Response 4: We have added correction about second step on Yun's citation (moved to line 132)
Point 5: the authors should include the research questions formulated in the Introduction, where they should detail the problem or gap that has been detected and on which a prediction has been made that later, in the methodology, has been corroborated or not. In addition, authors are asked to bear in mind that the research questions should be based on the literature review.
Response 5: We have added detail about research questions formulated in the Introduction the problem or gap that has been detected on the introduction section (line 70-87)
Point 6: In Methodology the sample and data collection is well explained, however, authors are asked to reflect the data in Figure 2 in the text and to include this Figure after the paragraph where it is mentioned and not before.
Response 6: We have added an explanation that reflects the data in Figure 2 in the text (line 271-281).
Point 7: The author should also justify the choice of questions in the questionnaire. Authors also have to include a table with statistics in the analysis of the results such as Reyes-Menendez, A., Palos-Sanchez, P. R., Saura, J. R., & Martin-Velicia, F. (2018). Understanding the influence of wireless communications and Wi-Fi access on customer loyalty: a behavioral model system. Wireless Communications and Mobile Computing, 2018.
Response 7: Considering that this study used QFD, which is a mixed methods analysis, we justify the selection of questions with reliability test (Cronbach's alpha) and validity test (Pearson's r correlation coefficient) (line 326-330). In addition, this study was also validated by triangulation of analysis methods, using affinity diagrams, tree diagrams and L-shaped diagrams as triangulated methods in terms of qualitative validity for the 5 main steps in the implementation of the quality function deployment method described in this study. We see that the 2 papers you suggested "Reyes-Menendez, A., Palos-Sanchez, PR, Saura, JR, & Martin-Velicia, F. (2018). Understanding the influence of wireless communications and Wi-Fi access on customer loyalty: a behavioral model system. Wireless Communications and Mobile Computing, 2018." and "Reyes-Menendez, A., Saura, JR, & Martinez-Navalon, JG (2019). The impact of e-WOM on hotels management reputation: exploring tripadvisor review credibility with the ELM model. IEEE Access, 7, 68868- 68877" are 2 outstanding and still closely related papers regarding understanding customer desires, and customer satisfaction. so we decided to cite these 2 papers in the literature review section to strengthen the theoretical basis of this paper (line 160-167).
Point 8: Response 8: We have added a description of the theoretical model on which this methodology is based (lines 332-351). In addition, the authors are asked to elaborate on the theoretical models on which this methodology has been based, such as those carried out by the authors Reyes-Menendez, A., Saura, J. R., & Martinez-Navalon, J. G. (2019). The impact of e-WOM on hotels management reputation: exploring tripadvisor review credibility with the ELM model. IEEE Access, 7, 68868-68877.
Response 8: We have added a description of the theoretical model on which this methodology is based (lines 332-351).
Point 9: For example in point 3.2, the QFD method should be referenced, as should the statement in point 4.1 when it says "Affinity diagrams are used extensively (...)".
Response 9: References on QFD (line 332-352) and affinity diagrams have been added (line197-197, and line 369-370)
Point 10: Although the results are correct, it is suggested that the author present them in a clearer way. For example, presenting on one side the research question, the result and whether or not it has been fulfilled.
Response 10: Please check line 534-566 for explanation by presenting the answer on one side the research question, the result.
Point 11: the justification of the formula used in this same point, 4.1, (line 342) or in line 431 of section 4.3. Regarding section 4.3, the data should be represented and explained beyond just showing the formula.
Response 11: We have added a data calculation process to explain the formulas we present (line 390-405, line 417-432, and line 525-529).
Point 12: In the results, the author is also asked to include a brief summary of the main conclusions drawn from the results.
Response 12: we have added a summary table for the conclusion (line 713-716).
Point 13: The results are correctly explained, but need the above-mentioned improvement.
Response 13: We have made some improvement, please check explanation previous response above.
Point 14: In the Discussion section the authors should include limitations and future research (points 6.2 and 6.3).
Response 14: We have included statements regarding limitations and future research in the discussion
Point 15: The Conclusion section is intended to clarify the objectives of the research, to state the aim of the research and to show what the authors are demonstrating with their research. Authors are asked to develop this section accordingly.
Response 15: We have modify the discussion and conclusion section (line 684-697).

Round 2
Reviewer 1 Report
THe authoes inserted my suggestions. I believe that the paper was improved.
Author Response
Thank you very much.
Reviewer 3 Report
The authors have properly addressed my suggestions
Author Response
Thank you very much